# CUDA-Optimized real-time rendering of a Foveated Visual System

**Elian Malkin, Arturo Deza†, Tomaso Poggio†***
Center for Brains, Minds and Machines (CBMM)
Massachusetts Institute of Technology
{emalkin,deza}@mit.edu;tp@ai.mit.edu

## Abstract

The spatially-varying field of the human visual system has recently received a resurgence of interest with the development of virtual reality (VR) and neural networks. The computational demands of high resolution rendering desired for VR can be offset by savings in the periphery [16], while neural networks trained with foveated input have shown perceptual gains in i.i.d and o.o.d generalization [27, 6]. In this paper, we present a technique that exploits the CUDA GPU architecture to efficiently generate Gaussian-based foveated images at high definition (1920px × 1080px) in real-time (165 Hz), with a larger number of pooling regions than previous Gaussian-based foveation algorithms by several orders of magnitude [10, 27], producing a smoothly foveated image that requires no further blending or stitching, and that can be well fit for any contrast sensitivity function. The approach described can be adapted from Gaussian blurring to any eccentricity-dependent image processing and our algorithm can meet demand for experimentation to evaluate the role of spatially-varying processing across biological and artificial agents, so that foveation can be added easily on top of existing systems rather than forcing their redesign ("emulated foveated renderer" [24]). Altogether, this paper demonstrates how a GPU, with a CUDA block-wise architecture, can be employed for radially-variant rendering, with opportunities for more complex post-processing to ensure a metameric foveation scheme [35]. Our code can be found here [21].

## 1   Introduction

Understanding the computational and representational goal of the foveated nature of the human visual field is a theme of ongoing research [25, 19, 11, 29, 5, 12]. Over the last decade, several works have suggested foveation as trying to achieve a *computational* goal tailored towards efficiency. These works support the previous hypothesis by proposing simulated experiments where a foveated perceptual system can achieve the same asympototic performance as a non-foveated perceptual system without any beneficial implications for a standard perceptual test such as object recognition (Akbas & Eckstein [2]). On the other hand, another current of thought has recently tried to find support for the *representational* goal of foveation (Rosenholtz [30]). Works of this flavor generally have shown that under specific perceptual tests, a foveated perceptual system can achieve an advantage over non-foveated perceptual systems by learning distinct representational signatures leading to robustness to scene occlusion (Deza & Konkle [6]), or reduction of false alarm rates in object detection (Pramod et al. [27]).

However, all previous works have approached this problem by studying a *stationary* visual agent that does not interact with the environment – a critical component suggested in developmental and perceptual psychology [13, 14, 34] that has slowly permeated into robotic systems [9, 32, 15, 18].

---

*Observations: First author is an undergraduate; †:Denotes shared senior authorship.

2nd Workshop on Shared Visual Representations in Human and Machine Intelligence (SVRHM), NeurIPS 2020.

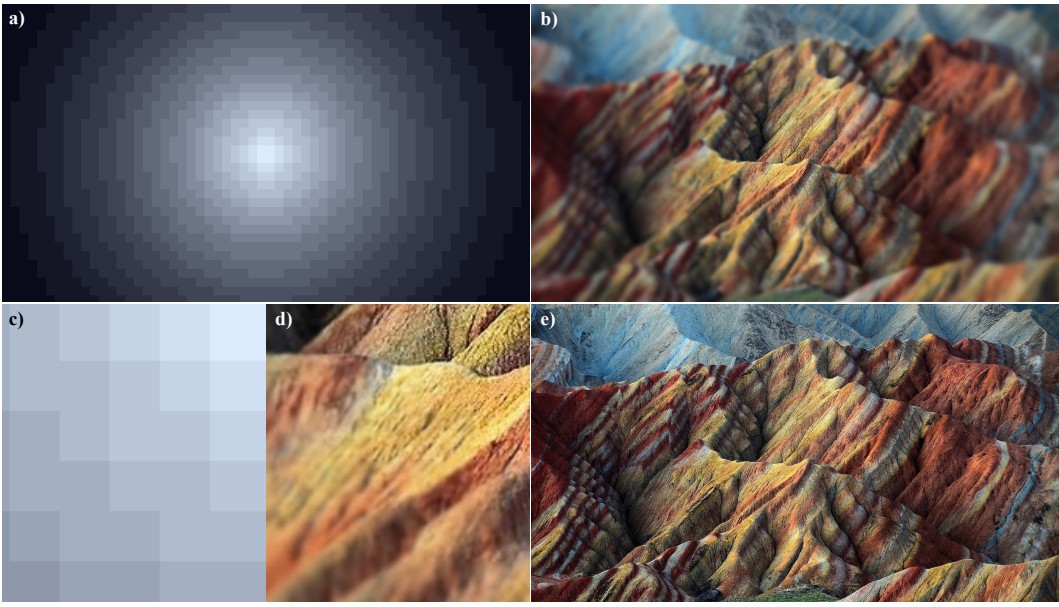

Figure 1: **a):** A $1920 \times 1080$ (PNG) image divided into $32 \times 32$ pixel blocks, each colored according to its distance from the fixation point (image center). **b):** Foveated $1920 \times 1080$ image, computed with our foveated optimization scheme based on adaptive gaussian blurring (generated at a rate of 165 Hz) **c):** Close up of image *a)* at transition from fovea into periphery. **d):** Corresponding close up of image *b)*. **e):** Original image. Notice that even without image blending such as cosine windowing functions, the final foveated output from *b) looks* smoothly generated in reference to *e)*.

Thus anticipating the advent of robotic agents that may require a foveated visual system to explore and interact with the environment (Orhan et al. [23]), – it will be critical to develop real-time efficient foveated image transforms if we are to study the perceptual implications of such a set of transforms (Deza et al. [7], Fridman et al. [8], Kaplanyan et al. [16]). Without such rendering speed, it will be computationally intractable to train reinforcement learning (RL) agents that require millions of trials – thus image frames – for training; or to deploy spatially-varying transformations on actual robotic agents such as drones that rely on portable GPUs.

In this paper we will address this issue by developing a neuro-physiologically inspired CUDA optimized foveated visual transform – that is based on adaptive gaussian blurring [10], which emulates convergence of retinal ganglion cells (RGC) as a function of retinal eccentricity in the visual field of a model system (Baden et al. [3]). Thus as we move one step forward in the direction of high-speed foveated rendering, the two-fold goal of this algorithm is: 1) To enable computational experiments that evaluate the role of spatially-varying processing via gaussian-like computation [27] in the periphery (RGC convergence) across biological and artificial agents [26, 3, 23, 33]; 2) To aid in the computational rendering tractability of next-generation Augmented/Virtual Reality (AR/VR) systems and robots (*i.e.* drones) that require adaptive low-energy powered systems [28, 37].

## 2   Proposed Approach: Block-wise Foveation

Block-wise foveation exploits the CUDA GPU architecture to efficiently generate foveated images with no subsampling and independently of the number of pooling regions. Our approach can be summarized in Figure 1, where the final image is assembled from square fragments, each of which has been blurred according to the distance from its midpoint to the point of fixation (that can be displaced anywhere in the visual field). To achieve this, each fragment in the final image is produced by a *single* CUDA block, with each block capable of using a different Gaussian filter to produce its uniquely assigned set of pixels. Given that the complexity is independent of the $(n)$-number of

pooling regions; $n$ can be much larger than previous foveation algorithms that are Gaussian-based by several orders of magnitude [10, 27], producing a smoothly foveated image that requires no further blending or stitching, and that can be fit for any contrast sensitivity function. By eliminating the need to subsample and interpolate we produce a higher quality adaptive blur transform with virtually no artifacts.

Foveating in blocks takes advantage of the small amount of shared memory available and specific to each CUDA block. Blurring is performed via separable convolution which is done in two stages - horizontal and vertical (See Appendix 5.1 – as a similar strategy is employed in the Gaussian Pyramid [4] and Feature Congestion [31]). The output of the first stage – the intermediate result – can be stored within shared memory, where only the threads *within* the block can access it. In doing so, the blocks work independently of each other to produce blurred fragments, making it simple to distribute blurring strengths across blocks. Figure 2 shows the wide range of Gaussian filters used to mimic the spatial frequency cut-off rates of the human retina – our algorithm concurrently employs all filters to process the image in a *single pass* (Algorithm 1).

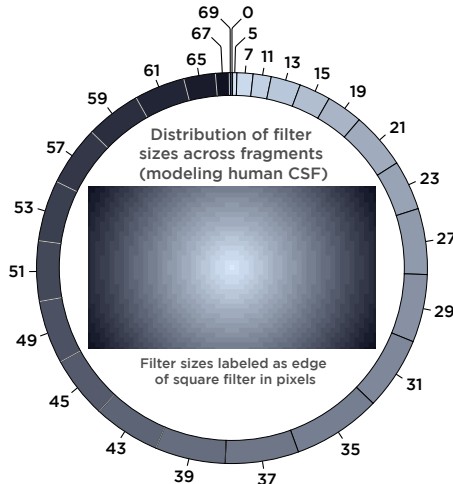

Figure 2: The filter sizes used to generate the foveated image in Figure 1, labelled in proportion to their representation in the image. The fragment patchwork has a total of 26 pooling regions. The CUDA kernel executes in 1.65 ms on an NVIDIA GTX 1060 GPU.

---

**Algorithm 1** CUDA Optimization Scheme for Foveated Rendering (See Appendix for detail)

  1: Choose fragment size ▷ **[CPU - Initial Setup:]**
  2: Define grid of blurring strengths $blur\ grid$ using retinal map or contrast sensitivity function
  3: Transfer array of Gaussian filters and array of cumulative sizes to GPU constant memory
  4: Transfer $blur\ grid$ to GPU global memory
  5:   Designate fixation point $f$ ▷ **CPU - With every frame: [Asynchronous]**
  6:   Calculate $blur\ grid$ offset to center on $f$, and $fragment\ shift$ to minimize fovea
  7:   Transfer input image from CPU pinned memory to GPU global memory
  8:   Place padded fragment into shared memory ▷ **GPU - With every frame: [Asynchronous]**
  9:   Perform horizontal convolution on padded fragment data
 10:   Place result of horizontal convolution into shared memory
 11:   Perform vertical convolution on horizontal convolution data
 12:   Write result to output array
 13: Transfer output image to CPU pinned memory

---

# 3  Experiments

To quantify the efficiency and precision of our foveated rendering scheme we devised a set of 2 experiments that assess both the rendering precision in addition to the theoretical-practical compute time along several hyper-parameters that are systematically varied.

## 3.1  Localized Image Quality Assessment

The Structural Similarity Index Measure (SSIM) is typically used to evaluate the quality of a compressed image with reference to the original [38]. We employ this metric to assess the accuracy of our implementation, using 'perfect' (loss-less) pixel-wise foveation as reference. Block-wise foveation scores lowest (0.971) near the fixation point, as can be seen in Figure 3. The spatial frequency function is steepest at the fovea (See Appendix 5.2.1), so relatively large differences arise at the edges of fragments, where the image is either over-blurred or under-blurred (blurring strength is set appropriate to the center of each fragment). To mitigate this effect, we shift the tiling of fragments to center a single fragment on the fixation point. This ensures that the untouched foveal region is as small as possible, minimizing the very first blurring step where the spatial frequency function is at its

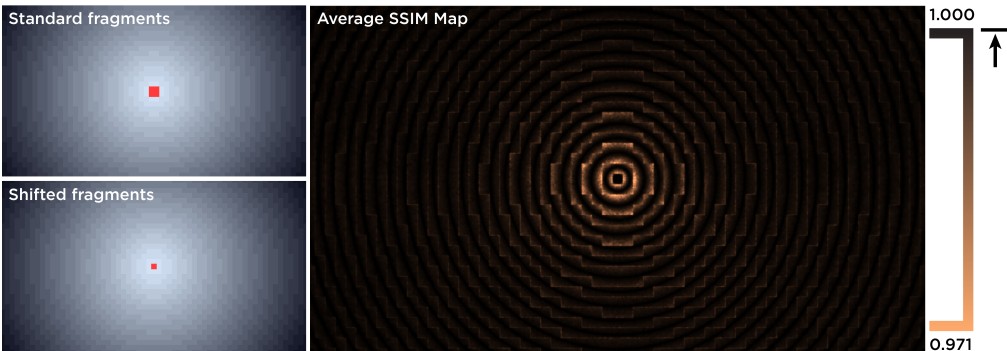

Figure 3: Average SSIM map was computed from 30 images including both natural, artificial, and abstract scenes. SSIM results show algorithm produces foveated images nearly identical to perfect foveation, with worst-performing regions scoring 0.971. Disparity is greatest near the fixation point where the spatial frequency drop off is steepest - shifting fragments to minimize foveal area (red) reduces disparity. Reference SSIM images were perfectly foveated on a single-pixel basis in 550 ms.

steepest. By tracking the center of the fixation point, the shift also serves to smooth the translation of the blur across the frame, whereas fixed tiling would force the blur to snap into place when the fixation point crosses the boundary of a fragment.

The ring-like pattern of under-blurring and over-blurring can be seen to continue out into the periphery with decreasing disparity as the function gradient becomes shallower. Although the edges of the fragments, where the steps occur, are imperceptible on still frames (Figure 1), they can become faintly visible with strong foveation and a large fragment size, as can be seen in the accompaning video [20]. Reducing the size of the fragments eliminates these artifacts. Moreover, it may be the case that these artifacts would not be perceptible to a user looking at the fixation point, though this would need to be evaluated through psychophysical experimentation.

## 3.2   Computation Time

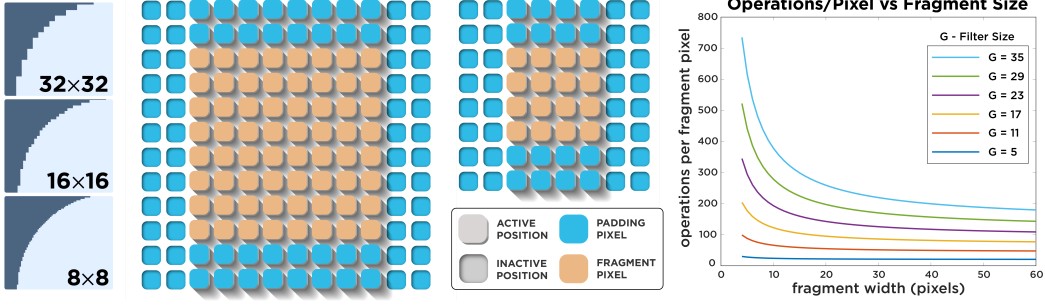

Figure 4: Fragment size (marked in pixel units) determines the smoothness of the boundaries between pooling regions, however, smoother boundaries are slower to compute. Larger fragments require fewer operations per output pixel. Fragment pixels are padded for the convolution operation - *e.g.* positions at which horizontal convolution is performed are shown as *active* positions. The ratio of padding to fragment pixels is lower for larger fragments, meaning relatively fewer operations are performed in the padding, for a given filter size ($G$) in pixels.

The time to compute a foveated image depends on the size of the fragments used. Smaller fragments are desirable because they result in more circular boundaries between pooling regions (Figure 4). Moreover, the smaller the fragments, the more pooling regions can be fit into the frame of the image. This leads to a finer discretization of the foveation curve and hence a better approximation of smooth retinal blurring. Smaller fragments are, however, slower to compute. Since it is necessary to perform work in the padding around a fragment during separable convolution, larger fragments "distribute" this extra work across more output pixels (Figure 4).

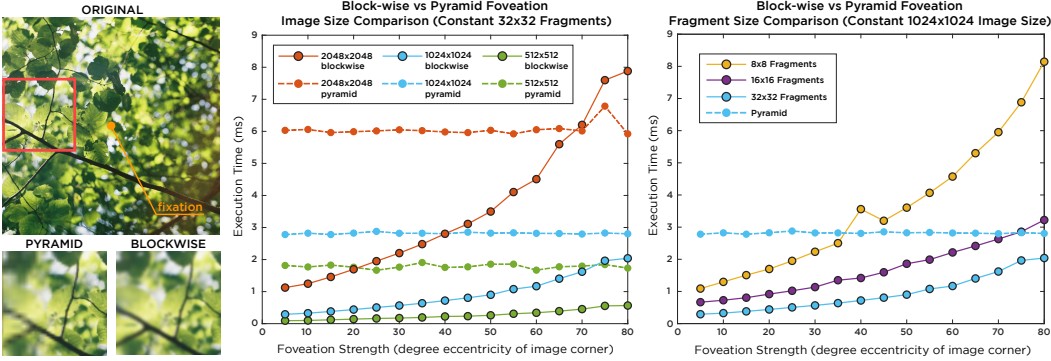

Figure 5: Evaluation is performed with RGB 8-bit images on a GTX 1060 GPU. For the block-wise approach, CUDA kernel execution is timed. For the Pyramid approach, mipmap generation and sampling is timed (See Appendix 5.1). Note that blue lines represent the same data in both figures.

In Figure 5, we compare the kernel execution time of our algorithm with the execution time of Gaussian pyramid foveation implemented in OpenGL [17]. The Gaussian pyramid approach generates a mipmap which is then used to sample or interpolate output pixels (Appendix 5.1). Interpolation tends to produce blocky results, as compared to the smooth blur of our convolutional block-wise approach. The execution time of our algorithm rises with increasing foveation strength, which corresponds to larger Gaussian filters and hence more operations during convolution. The performance of the Gaussian pyramid approach only varies with image size, as for a given image size mipmap generation and subsequent sampling require the same number of operations across all foveation strengths.

While Figure 5 demonstrates timings for center-frame fixation, execution time also varies with the location of the gaze in the image. Fixating on the edge of the frame introduces larger filters on the opposite edge, when compared to center fixation, therefore increasing computation time. On top of kernel execution, data transfer between the CPU and the GPU can significantly affect the total time to generate a foveated image. Although the $1920 \times 1080$ px image in Figure 1 took just 1.65 ms to foveate, the time spent in transfer was 4.4 ms (using pinned memory), leading to a foveation rate of 165 Hz. In general, several steps can be taken to further reduce the total foveation time, if more complicated foveation structures are required (See Discussion 4). Execution time can be accelerated by using half-precision floating-point arithmetic, as well as more powerful GPUs. The impact of data transfer on real-time performance can be mitigated by overlapping kernel execution with transfers to and from the GPU. Figure 6 showcases other CUDA-Driven optimization strategies that are discussed more in depth in Appendix 5.5.

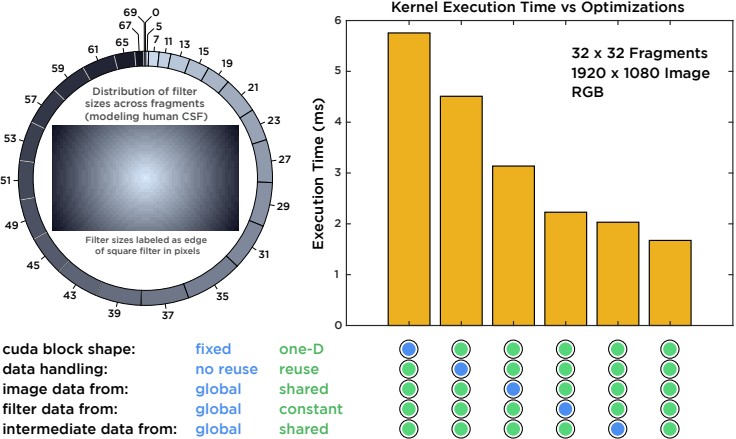

Figure 6: Impact of optimizations on performance. Evaluation done on NVIDIA GTX 1060 GPU (6 GB). Only kernel execution times are shown - memory transfers excluded.

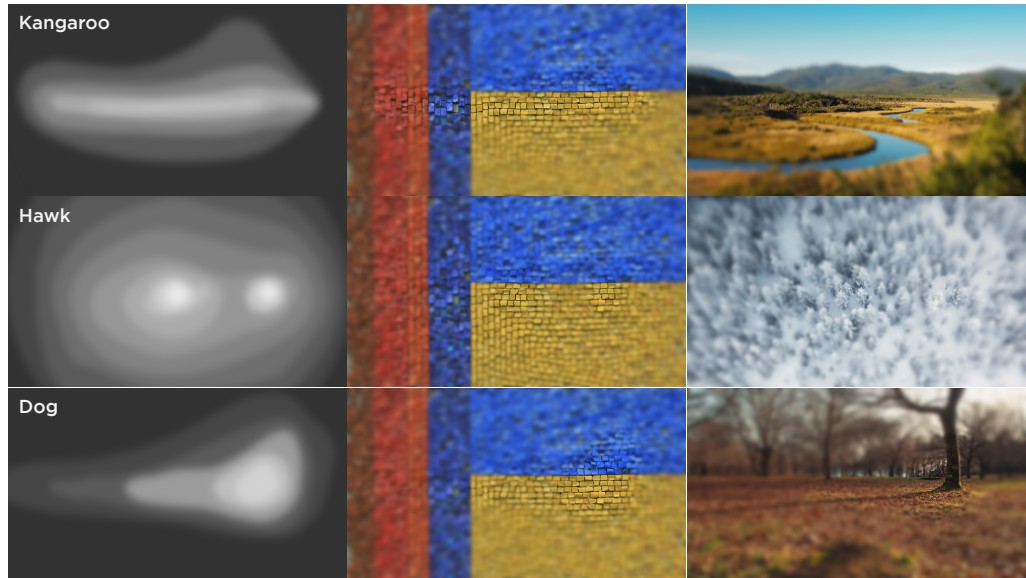

Figure 7: Sample retinal ganglion cell (RGC) density distributions across retinas of different species. Left: the smoothed RGC maps of the kangaroo, hawk, and dog are shown (sketched from Baden et al. [3]). Each map is used to asymmetrically foveate an image in *real-time* of a mosaic (Middle) and an image of the potential environment of the relevant animal (Right). Our rendering pipeline opens the door towards training RL agents and CNNs on visual input of spatially-varying properties.

## 4 Discussion

The flexibility of our block-wise algorithm means that it can be extended to non-radially-symmetric functions with no additional computational overhead. *Any* image of the shape $f(x, y) : \mathbb{R} \times \mathbb{R} \to \mathbb{R}$ can be used to describe blurring strength throughout the rendering frame, such that the resultant foveation can represent any desired retinal structure. Figure 7 shows an example of how our rendering scheme can be adapted to a variety of animal retinas given their retinal ganglion cell (RGC) convergence maps. These images allow us to probe the use of different retinal structures through reverse-engineering of perceptual architectures, where we can train different machine vision systems with asymmetric image processing schemes to evaluate how well they may learn a task such as object or scene classification. Naturally, each RGC map is usually coupled with its own set of image statistics such as colors, vantage point and structure.

Altogether, rendering stimuli as viewed from a biological agent may aid us in understanding the development of their visual system. In return, given that our framework is real-time, it opens the door to test new theories of visual perception via Reinforcement Learning (RL) agents that may require millions of visual training frames that need not be spatially uniform, and that can later be deployed in robots (Figure 8).

Finally, the rise of several AR/VR systems that have human users (who have foveated visual systems) may benefit from accelerated rendering systems as well [16]. Whether it is for purely entertainment purposes (video games), Human-Machine Interfaces such as tele-operation, or VR-driven psychophysics, the high demand of real-time foveated rendering procedures may become useful in the near future.

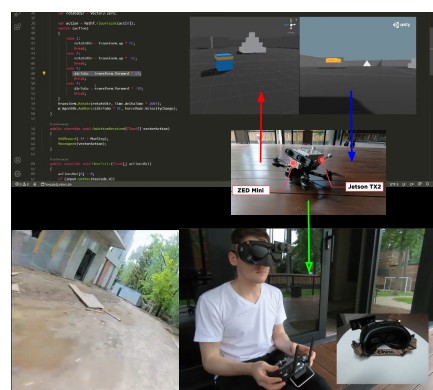

Figure 8: Future applications of real-time foveated rendering may enable training RL Agents that learn with foveated visual systems in virtual environments, to later deploy them into drones for automatic or semi-automatic (VR-assisted) navigation.

## Acknowledgments

Authors would like to thank MIT's Center for Brains, Minds and Machines (CBMM), the National Science Foundation (NSF), MIT Quest for Intelligence, and Lockheed Martin. Authors would also like to thank Ronald Alvarez for permissions to use the upper portion of Figure 8, and members of the Poggio Lab for insightful comments during lab discussions.

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

# 5   Appendix

## 5.1   Previous Work on Image Foveation

Many techniques have been proposed in the domain of computer graphics to introduce foveation into the rendering pipeline in order to reduce computational cost. Most similar in design to our algorithm is Variable Rate Shading (VRS) - an NVIDIA Turing GPU feature that tiles the image frame and adapts shading rate from tile to tile to increase rendering performance versus individual pixel shading [22]. This work differs from ours by targeting the shading step; our algorithm is proposed as a post-process and applied to the existing full-resolution image to enable experimentation with foveated visual systems under the requirement of real-time performance.

Previous work on image foveation as a post-process has employed Gaussian pyramids *a la* Burt & Adelson [4] to generate foveated images at reasonable time scales [10]. This technique convolves an image with a small Gaussian filter, followed by subsampling of the pixels, reducing the size of the image by a factor of four while avoiding aliasing issues thanks to the preceding low-pass convolution. Repeating these steps yields a series of sequentially smaller images that can be imagined to make up a pyramid. Upscaling each image back to its original size produces increasingly blurrier versions of the original, which can then be combined into a single foveated image. Mipmaps are used in computer graphics and employ the same technique to render scaled-down textures without aliasing. The rendered pixels are interpolated from the mipmap as a fractional level of the pyramid, i.e. from between two pyramid levels.

The foveation quality of the Gaussian pyramid approach suffers from its reliance on interpolation, which produces a blocky image. Bilinear interpolation is frequently used as a middle ground between the pixelated results of nearest neighbor sampling and the high computational cost of bicubic interpolation, however, it cannot provide the same smoothness as a purely convolutional blur scheme.

## 5.2   Methods

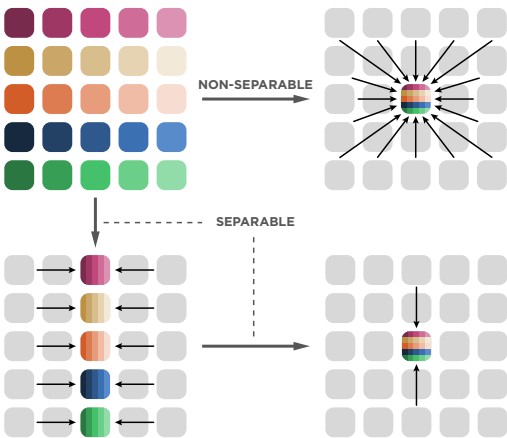

Figure 9: Separable convolution is faster than non-separable convolution. Although the separable approach requires more operations to produce a single output pixel, the intermediate horizontal result is re-used for the vertical step by several pixels, leading to a net decrease in the number of operations.

To render foveated images, it is necessary to designate a fixation point and perform increasing blur into the periphery. Blur can be achieved by convolving the image with a filter – a small matrix of values that is centered on every pixel in the image, followed by an element-wise multiplication and sum. Although several filters exist for achieving a blur effect, we employ the Gaussian filter, owing to its smooth blur and lack of ringing effects, as well as to its separable property that reduces computational complexity.

The computational demands of the convolution operation can render it prohibitive for time-sensitive tasks. Gaussian blurring of an image of width $w$ and height $h$ with a filter of dimensions $N \times N$ has a complexity of $O(whN^2)$. Consequently, high quality foveated rendering, with large peripheral averaging filters and high resolution images, presents a challenge by virtue of the number of operations it requires. A key step in lowering convolution complexity is taking advantage of the separable property of the Gaussian filter, reducing it from an $N^2$ square filter into two $N$-length one-dimensional ($1D$) filters [4]. The convolution of the two-dimensional square filter is equivalent to first a horizontal convolution over the image with the first filter, and then a vertical convolution with the second over the result of the horizontal pass. In the case of a Gaussian filter both $1D$ filters are identical, hence only one needs to be stored for both the horizontal and vertical step. Separable convolution is significantly faster by reducing the number of operations such that the computation can be performed in $O(whN)$ time.

The Gaussian filter is defined by its standard deviation $\sigma$, with a larger $\sigma$ producing a larger spread of values, and hence more averaging and a stronger blur.

$$G(x,y) = \frac{1}{2\pi\sigma^2} e^{-\frac{x^2+y^2}{2\sigma^2}} \tag{1}$$

Although the Gaussian function decays out to infinity, we must determine a cutoff point for our filter to use during convolution. Values beyond $3\sigma$ from the center of the filter are small enough to be considered negligible, hence each filter is dimensioned as $\lceil 6\sigma \rceil \times \lceil 6\sigma \rceil$. The filter matrix is populated with the output of the Gaussian function at each discrete matrix coordinate (center pixel - (0,0)), then normalized to prevent the convolution operation from altering the luminance of the image.

### 5.2.1 Choosing the Standard Deviation of the Gaussian Filter

The standard deviation of the Gaussian filter used at a given eccentricity is found through the method used by Geisler and Perry [10], starting with the contrast threshold formula, which gives the contrast required for detection of a patch of sinusoidal grating of a given spatial frequency $f$ (cycles per degree) at retinal eccentricity $e$ (degrees):

$$CT(f,e) = CT_0 \exp\left(\alpha f \frac{e + e_2}{e_2}\right) \tag{2}$$

where $CT_0$ is the minimum contrast threshold, $\alpha$ is the spatial frequency decay constant, and $e_2$ is the half-resolution eccentricity. We set these values to be the same as used by Geisler and Perry [10], such that $\alpha = 0.106$, $e_2 = 2.3$, and $CT_0 = \frac{1}{64}$. Setting the left-hand-side of the equation to 1.0, which denotes maximum contrast, and rearranging for $f$ gives:

$$f_{c\_deg} = \frac{e_2}{\alpha(e + e_2)} \ln \frac{1}{CT_0} \tag{3}$$

where $f_{c\_deg}$ is the highest spatial frequency (cycles per degree) that can be resolved at eccentricity $e$, no matter the contrast. This equation therefore gives the spatial frequency cutoff at a given retinal eccentricity.

In a metameric foveation scheme the viewing distance and monitor dot-pitch are taken into account to calculate retinal eccentricity and create foveated images indistinguishable from non-foveated images under the set conditions (e.g. Pramod et al. [27]). Instead, we use a retinal mapping approach, where the blurring factor scales proportionally to the falloff of visual acuity of the human retina. The maximum spatial frequency in an image is a single cycle occupying two pixels, i.e. a black and white

pixel in succession, which equates to 0.5 cycles per pixel. We use the following equation to derive the spatial frequency in cycles per pixel from the spatial frequency in cycles per degree:

$$f_{c\_pix} = 0.5 \cdot \frac{f_{c\_deg}}{f_{max}} \tag{4}$$

where $f_{max}$ is the maximum resolvable spatial frequency across the retina, corresponding to highest visual acuity at the fovea. The degree eccentricity of a pixel is calculated by first assigning the image center to $0°$, and the corner of the image to a desired retinal eccentricity. By interpreting the flat image as a spherical retina, we create a linear relationship between Euclidean pixel distance and retinal eccentricity:

$$e = \frac{d_p}{d_{corner}} e_{corner} \tag{5}$$

where $d_p$ is the pixel distance from the fixation point, $d_{corner}$ is the half-diagonal image distance, and $e_{corner}$ is the degree retinal eccentricity assigned to the image corner. In this way, we produce foveated images designed to represent the human visual experience relative to the resolution of the original image, rather than metameric foveation under any specific conditions. The algorithm is simple to adapt to achieve the latter.

Finally, we find the standard deviation $\sigma$ of the Gaussian filter at the chosen eccentricity by treating $f_{c\_pix}$ as the standard deviation of the filter in the frequency domain, $\sigma_f$:

$$\sigma = \frac{1}{2\pi f_{c\_pix}} = \frac{1}{2\pi \sigma_f} \tag{6}$$

This signifies that frequencies at $\sigma_f$ and above are considered cut off, or sufficiently attenuated by the Gaussian filter. This threshold can be shifted higher for stronger blurring.

### 5.3 Localized Image Quality Assessment dataset

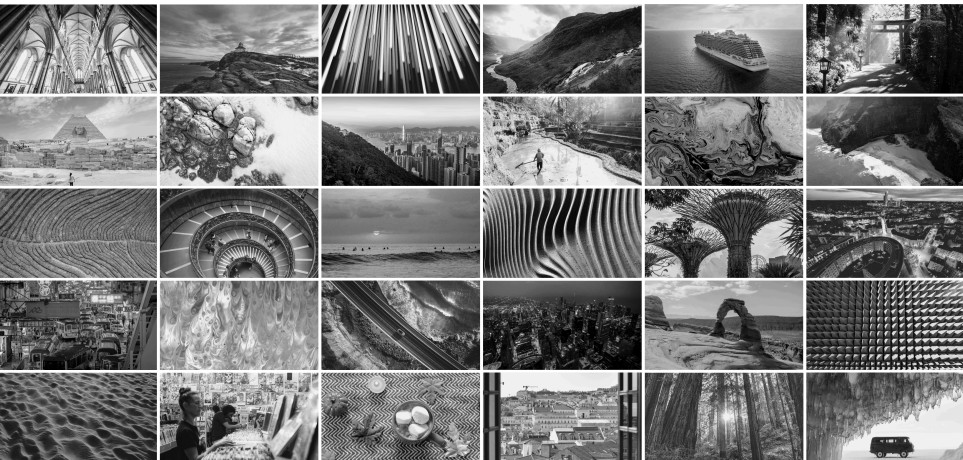

Figure 10: Images used (30) for calculating the average SSIM map of the block-wise algorithm with reference to perfect foveation (Section 3.1).

## 5.4 A primer on the CUDA Architecture

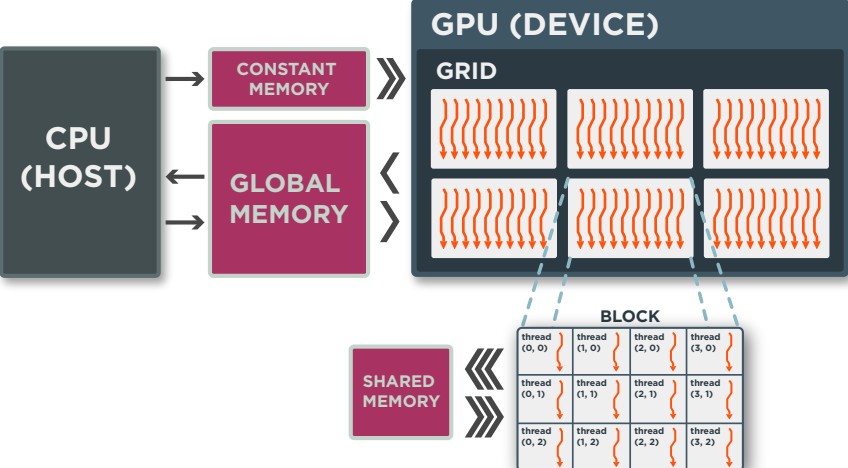

Figure 11: The three levels of the CUDA architecture. The GPU is divided into a grid of blocks, each of which holds a collection of threads. Each thread executes a single parallelized computation. Communication between the CPU and GPU is performed via global memory (slowest). Constant memory communication is one-way (faster). Shared memory is "shared" only by the threads within the same block (fastest).

CUDA is the programming interface through which custom programs (kernels) can be executed on an NVIDIA graphics processing unit (GPU). The advantage of employing a GPU for foveation is that a GPU is able to perform parallel computation of the same kind on large datasets, hence is an ideal candidate for a convolution operation involving millions of pixels. The GPU is used in tandem with the CPU to accelerate highly parallel operations. The CPU (host) handles the transfer of data to the GPU (device), which executes the kernel code written with CUDA.

Figure 11 shows the basic CUDA architecture, which consists of three levels – the threads, the blocks, and the grid. Each thread performs one of the parallelized computations, such as a single write of a blurred pixel to an output array. The CUDA threads are organized into blocks, which reflect the physical design of the GPU, where the workload is distributed across Streaming Multiprocessors. The threads within the same block have the useful property of being able to communicate with each other via a fast form of memory (shared memory), which is crucial to the optimization of CUDA code. The blocks are organized into the grid, which ties together the whole structure of the CUDA kernel. The grid's purpose is to define how many blocks a kernel is launched with, and how these blocks are arranged. Both the grid and the blocks can have 3 dimensions, which helps organize the programmer's solution [1].

### 5.4.1 The CUDA memory model

Optimization of device kernels requires understanding of the different types and properties of memory offered by CUDA.

Global memory is the most basic type of memory available on the GPU – it is the memory typically accessed by the host when data is transferred to the device. Global memory offers the largest storage size on the GPU, on the order of several gigabytes, but has the drawback of being slow to read from and write to (it is located off-chip, away from the streaming multiprocessors).

Constant memory is a faster type of memory. It is termed "constant" because it can only be written to by host code, hence is useful when the kernel requires access to read-only data. The size of this memory is far smaller than that of Global memory - only 64KB [1]. Although it is still off-chip, it is

nonetheless much faster to access than global memory because it is aggressively cached to on-chip memory.

Each thread block is designated a small amount of shared memory (48-163 KB) which can be written to and read from by that block alone. Shared memory is located on-chip and is therefore much faster to access, making it useful for storing intermediate values within a kernel, or data that must be accessed repeatedly.

The GPU architecture also features registers and local memory, which are specific to individual threads. Registers offer storage for any variables or arrays declared by a thread within the kernel, and are the fastest to access, but are limited in quantity. The CUDA compiler determines which data is placed within a register, and which data spills out into local memory – termed local not because of its location, but because of its scope. Local memory is off-chip but accessible only by the specific thread, hence it is local to each thread but slow to access. As a result, register spilling is highly undesirable.

## 5.5   CUDA-Driven Optimization

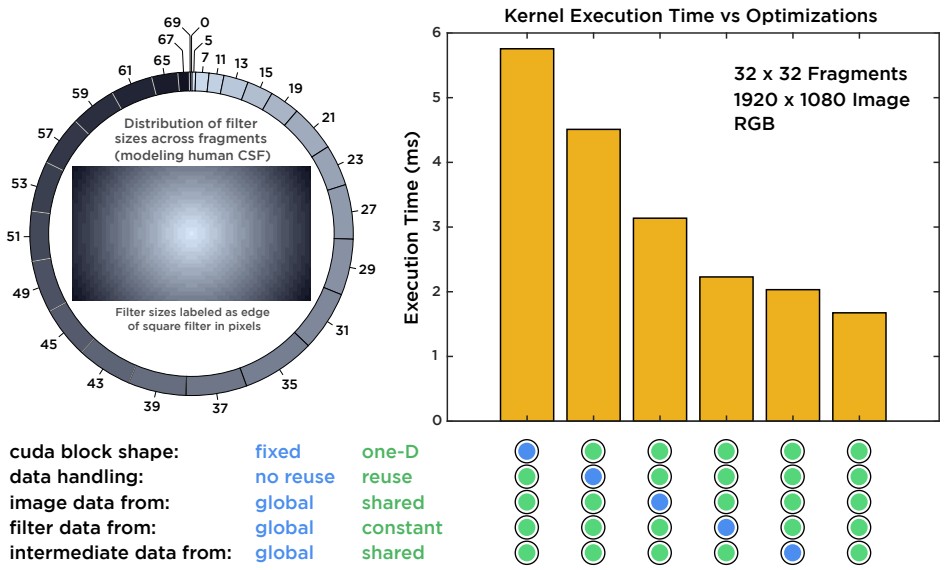

Figure 12: Impact of optimizations on performance. Evaluation done on NVIDIA GTX 1060 GPU (6 GB). Only kernel execution times are shown - memory transfers excluded.

Our algorithm demonstrates the suitability of CUDA's block architecture to space-variant image processing. We detail below the considerations of the kernel design such that the algorithm can be adapted to other foveation schemes while maintaining high performance.

### 5.5.1   Workload of the Block

Four arrays are placed on the GPU: the input image to be foveated, the *blur grid* describing the blurring strengths of each fragment, the array of one-dimensional Gaussian filters $a_g$, and an accompanying array of cumulative filter sizes $a_c$.

The threads within the block identify their position on the CUDA grid, which corresponds to a value in the *blur grid*. This value is an index into $a_c$, which allows the thread to extract the appropriate Gaussian filter from $a_g$. If the block is assigned the foveal fragment, the pixels from the input image are simply written directly to the output array - no blurring is done. Otherwise, the block performs Gaussian blurring on its assigned pixels using the extracted Gaussian filter.

The pixels corresponding to the fragment are padded according to the size of the Gaussian filter used. This *tile* of pixels is extracted, and horizontal convolution is performed. The result of this operation is the half-blurred data, termed the *intermediate result*. Convolution is performed a second time, with the same Gaussian filter, but this time in a vertical manner, producing the final blurred fragment. The result is written to the output array.

### 5.5.2   Shared Memory

Shared memory is faster than global memory because it is located on-chip, hence we can achieve higher performance by using it to hold frequently accessed data. Since the convolution operation accesses the same pixels multiple times, we can first load the *tile* of pixels from global memory into shared memory before the horizontal pass. This extra step leads to a performance boost by reducing latency throughout the convolution. We take care to retain the unsigned 8-bit integer image datatype until convolution, which minimizes the data transfer time and the storage footprint in shared memory.

The *intermediate result* is likewise placed into shared memory, as it is specific to the workload of the individual block and each thread must access the results of horizontal convolution generated by vertically neighboring threads. Therefore, the same benefits of shared memory are applied to the vertical convolution pass.

### 5.5.3   Constant Memory

The storage size of constant memory is too small to store the image to be foveated, but is sufficient to hold the Gaussian filters used for convolution. These filters remain unchanged throughout the kernel execution, but are frequently accessed, typically millions of times during foveation. Constant memory is therefore an ideal location because it offers low latency in exchange for read-only designation.

The two $1D$ filters produced by the separation of the $2D$ Gaussian filter contain identical values - their only difference is the direction in which they are applied. We therefore only store one instance of an $N$-length filter and apply it in a horizontal or vertical fashion as appropriate. All Gaussian filters are transferred together into array $a_g$ in constant memory, and their cumulative sizes are likewise placed into constant memory as a second array $a_c$. Cumulative filter sizes allow the kernel to locate the start and end point of any given Gaussian filter within $a_g$.

### 5.5.4   Optimizing for Cancelled Threads

Although image processing typically involves two-dimensional thread blocks to map individual threads to image pixels, this approach is inefficient for space-varying algorithms. There are three main stages in the execution of the kernel - pixel transfer into shared memory as a padded *tile*, followed by the horizontal and vertical convolution passes. These three steps each require a different arrangement of threads and vary with Gaussian filter size which determines the amount of padding around a fragment. Therefore, a two-dimensional thread block is impractical, as a fixed thread arrangement across the block workload would lead to unused threads at several points in the algorithm. Moreover, by launching blocks with fixed dimensions to accommodate the largest filters in the periphery, we would lose performance on smaller tiles closer to the fixation point.

Optimal performance for every task throughout the workload can be achieved by launching each block as a one-dimensional (*one-D*) line of threads. Within the kernel, the dimensions of active positions within the tile are specified at every step, so that the line of threads can be organized to best fit the task at hand. Figure 13 shows how a one-dimensional block of threads can be reshaped.

The number of active positions in the tile is usually greater than the number of invoked threads, so blocks typically iterate through the tile. The number of iterations is reduced for one-dimensional blocks as compared to fixed two-dimensional blocks as threads populate only active positions (with the exception of the last iteration where overflow might occur).

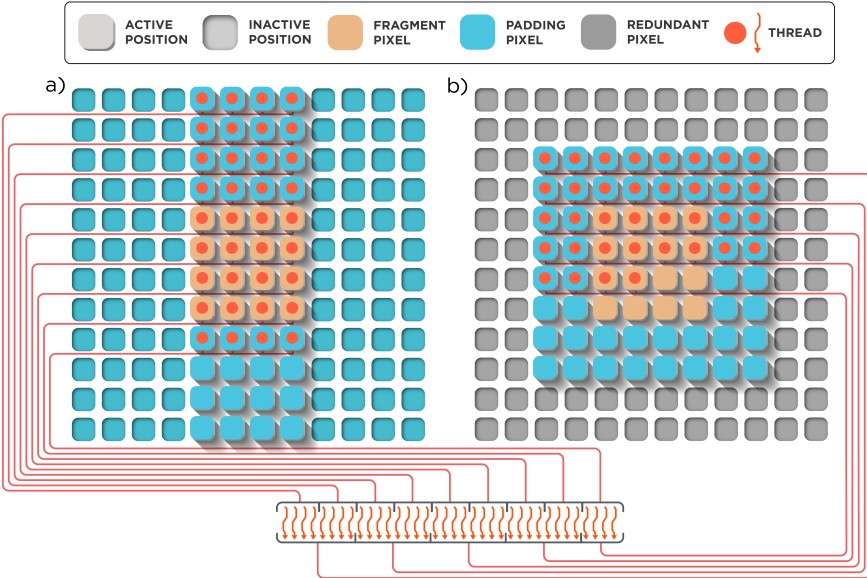

Figure 13: A one-dimensional block of threads can be optimally arranged for any purpose. **a)** Threads arranged in a vertical column for horizontal convolution. **b)** Threads arranged to transfer a small tile of image data into shared memory.

### 5.5.5 Data Reuse

An optimization discussed by Volkov [36] involves reusing data accessed within the kernel through registers, which reduces accesses of slower forms of memory. This technique can be applied during the convolution step of the algorithm.

During convolution, the same Gaussian filter is used in an overlapping fashion for many neighboring pixels. If the convolution operation at every pixel is assigned a separate thread, neighboring threads access the same image and filter data several times. Performance can be improved by increasing the work done by a single thread, such that memory accesses and data transfers are reduced. By having a single thread perform four convolutions at once, we achieve over 2x speedup (Figure 12). The exact factor by which to increase per-thread workload can be determined through experimental evaluation. Note that since there are typically many more pixel positions in the tile (at which convolution is to be performed) than there are invoked threads, increasing per-thread workload does not lead to inactive threads but rather reduces the number of iterations through the tile.

Data reuse involves registers - on-chip memory that holds variables and arrays declared by a thread. Registers are the fastest form of memory to access but are limited in quantity, therefore they can be used for temporary storage of values fetched from less efficient memory locations. Figure 14 shows the first two steps of horizontal convolution performed by a thread, where each step is concerned with a different Gaussian filter value. In Step 1, the first filter value is accessed. This value will be used as the first step of convolution for all four highlighted pixels, so it is placed in a register and applied immediately at the appropriate positions - four separate convolution sums are maintained. Reusing the filter value is faster than extracting it four times from constant memory. Step 2 uses three of the four image pixel values from the previous step, so these values are likewise reused, rather than being fetched anew from shared memory. The reuse pattern seen in Step 2 continues for the remaining steps.

### 5.5.6 Algorithm Limitations

The optimized algorithm is limited by the sizes of constant and shared memory on the GPU. On an NVIDIA compute capability 6.1 device, the maximum amount of shared memory per thread block is

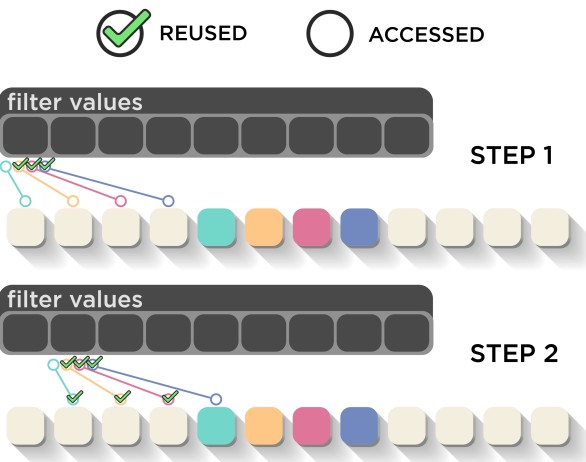

Figure 14: Data reuse through local variables improves performance of the kernel. The convolution operation has a great deal of overlap at neighboring pixels. A thread is able to store values in registers and perform four pixels worth of convolution, allowing a single access of constant or shared memory to serve multiple convolution operations. Step 1 shows reuse of the first filter value. In Step 2, the second filter value is likewise reused but so are three of the four image pixels previously accessed.

48 KB, and the constant memory size is 64 KB [1]. These limits impose constraints on the number of pooling regions and largest filter size that can be used by the algorithm.

Shared memory is used for image data and the *intermediate result* produced by horizontal convolution. The image data is in the form of a tile of width $tileW$ and height $tileH$, which holds fragment data of dimensions $fragmentW$ and $fragmentH$, as well as the surrounding padding. The tile is stored in shared memory with *unsigned 8-bit integer* datatype - the same as the input image. The array that holds the *intermediate result* has dimensions $fragmentW * tileH$ with *float32* datatype, having been produced through multiplication with floating point Gaussian filter values. Since *float32* values are four times the size of *unsigned 8-bit integers*, the *intermediate result*, despite holding fewer values, can take up more shared memory than image data. Shared memory usage can be calculated in bits as follows:

$$\text{Shared Memory Used} = tileW * tileH + 4 * (fragmentW * tileH) \tag{7}$$

Larger Gaussian filters require more padding around a fragment leading to a larger $tileW$ and $tileH$. For fragment dimensions of $32 \times 32$, $16 \times 16$, and $8 \times 8$ pixels, the maximum filter size is 133, 174, and 196 pixels respectively.

Gaussian filters and their cumulative sizes are placed into arrays $a_g$ and $a_c$ in constant memory. The number of bits placed into constant memory is a function of the total number $N_g$ of *float32* values in array $a_g$ (total length of all filters used), as well as the number $N_c$ of *int32* values in $a_c$ (number of filters used, i.e. number of pooling regions):

$$\text{Constant Memory Used} = 4 * (N_g + N_c) \tag{8}$$

$N_g$ is determined by the lengths of the Gaussian filters which correspond to blurring strength defined by specific foveation parameters. The maximum number of pooling regions, each of which is defined by a unique filter, must therefore be determined experimentally.

The limiting factors of shared and constant memory can be removed through the use of global memory in exchange for lower performance. The impact of these compromises on execution time is shown in Figure 12.

