# OpenReview forum: "CUDA-Optimized real-time rendering of a Foveated Visual System"
_NeurIPS.cc/2020/Workshop/SVRHM — SVRHM@NeurIPS Poster_

### Official Review · AnonReviewer1 · 2020-10-27
**An excellent contribution that aligns well with the goals of the workshop**

**Rating:** 9
**Confidence:** 4

**Review:**

This paper proposes "Block-wise foveation", an image foveation algorithm optimised for CUDA capable hardware. The authors show that their approach closely approximates perfect foveation and that it is favourable to a simple pyramid approach both in terms of execution time and the smoothness of the result. Finally, the authors demonstrate that arbitrary images can be used to describe the blurring strength such that foveation of any shape can be considered. The paper is an excellent contribution that aligns well with the goals of the workshop. That said, the authors may wish to address the following:
- The paper draws heavily on ganglion cell distributions as the biological analogue of image foveation. However, in addition to foveation, ganglion cells perform a function commonly seen as feature extraction. Image foveation is therefor perhaps better described as modelling the spatial non-uniformity (in size and density) of photo-receptors. It might be of value to include some mention of this caveat in the paper.
- There are several settings in which it may be of value to learn the parameters of the fixation point or even the blur strength image. Have the authors considered any schemes through which one might obtain a gradient with respect to the foveation inputs / parameters? At the very least it _may_ be possible to consider a pure PyTorch implementation which (although likely much slower) permits automatic differentiation.
- The algorithm is clearly non-trivial to implement, particularly for anyone who is not familiar with the CUDA GPU architecture. It would be of value for the authors to consider releasing their code upon de-anonimysation of the paper.

Minor typo
Line 37. "of such transform" -> of foveation / of such __a__ transform?

---

### Official Review · AnonReviewer3 · 2020-10-30
**Simple implementation of foveation on CUDA parallel hardware**

**Rating:** 6
**Confidence:** 3

**Review:**

The paper discusses a strategy to compute a foveated blurring of images particularly on CUDA-enabled hardware.

I enjoyed the discussion of using a Gaussian pyramid vs. a block-based strategy for implementing eccentricity-dependent Gaussian blurring. However, given that this is a signal processing-heavy topic, I would have expected a more precise (i.e. mathematical) description of the algorithm (and perhaps, for comparison and completeness, the algorithm based on Gaussian pyramids). This, and a description of the overall role of Gaussian blurring in foveated computation would have been very helpful to make this a self-contained paper. It is still somewhat unclear to me what the exact role of the blurring is in the implementation of pooling regions. They are mentioned, but the exact computational model of pooling that the authors work with is never defined.

With respect to the signal processing aspects of the paper, I find it disappointing that the authors resort to a mathematically somewhat unambitious formulation. It is clear why the algorithm doesn’t scale well with greater foveation strength: It simply needs to compute responses of larger and larger kernels in the periphery. While the authors exploit the separability of 2D Gaussian kernels into two 1D kernels, the Gaussian pyramid can exploit this, and also the fact that the composition of two Gaussian kernels is equivalent to one Gaussian kernel with larger variance. Furthermore, it makes use of the Nyquist theorem, stating that sufficiently blurred signals can be downsampled while still being near perfectly reconstructible using linear filtering. The “blockiness” that the authors report when using the Gaussian pyramid approach hence must be due to the way interpolation between the scales of the pyramid is performed. Perhaps with a more sophisticated interpolation scheme, better results could be obtained, while avoiding block-based processing?

Lastly, I may be missing some context here, but again, since the role of Gaussian blurring in the context of generic foveated computation isn’t clear to me (other than as a model for retinal acuity), I wonder how universal the results of this paper are for models of foveated processing further down the visual hierarchy. For instance, is it still permissible to assume separability for cortical processing stages?

---

### Official Review · AnonReviewer2 · 2020-10-31
**Reasonable tutorial for GPU optimization of foveation filters for real-time gaze-contingent systems**

**Rating:** 6
**Confidence:** 3

**Review:**

This work describes a GPU-optimized method for a high-performance computation of foveated filters of arbitrary shapes. While the paper applies a fairly common set of GPU optimizations, it can be useful for the community both as a fast tool or benchmark implementation, as well as a tutorial on leveraging GPU for image processing tasks. The paper does a fairly good job at comparing to the existing implementation as well as providing the ablation study of applied optimizations.

I have a few comments/suggestions:
- The authors claim in the abstract the real-time performance of 165Hz. It would be good to clarify that this number depends on various factors, such as the location of the gaze in the image and the amount of foveation. Also, as a nitpick, I would recommend the authors to use milliseconds instead of Hz to report real-time performance, since it's more linear and also easy to add up with other latencies of a gaze-contingent system.
- My major concern with this approach is around the fixed and sharp boundaries of the tiles. While the authors demonstrate that the tile boundaries are unnoticeable in figures, I would argue that these boundaries can become noticeable artifacts in motion, especially when dynamic content and smooth pursuit are considered. I would highly recommend the authors to evaluate it in a supplementary video and add the description in the text.
- Last but not least, I would highly recommend the authors to make their implementation open source for the good of the community.

---

### Public Comment · ~Elian_Malkin1 · 2020-12-11
**Response to Reviewers**

Dear Reviewers,

Thank you very much for your feedback and helpful comments. We have incorporated many of your suggestions into the camera ready version of the paper, including a link to a GitHub repository with the code files. Another supplementary addition is a demonstration posted on YouTube, which aims to address the concern about visible tile boundaries in motion. Thank you again for your thoughtful feedback!

Sincerely,
The Authors

---

### Decision · Program_Chairs · 2020-11-02

Accept (Poster)